# Implementation evaluation of a medical student-led intervention to enhance students' engagement with research: Findings and lessons learned

**Mian Arsam Haroon**[1]*, **Ali Aahil Noorali**[2], **Abdullah Saeed Khan**[1], **Muzamil Hamid Hussain**[1], **Rohan Advani**[1], **Ashmal Sami**[1], **Asma Altaf Merchant**[1], **Adnan Ali Khan**[1], **Sana Gul Baloch**[3], **Arsal Tharwani**[4], **Saulat H. Fatimi**[5], **Zainab Samad**[2], **Babar S. Hasan**[6], **Muneera A. Rasheed**[6]

1 Medical College, Aga Khan University, Karachi, Pakistan, 2 Department of Medicine, Aga Khan University, Karachi, Pakistan, 3 School of Nursing and Midwifery, Aga Khan University, Karachi, Pakistan, 4 Department of Medicine, Cleveland Clinic, Cleveland, OH, United States of America, 5 Department of Surgery, Aga Khan University, Karachi, Pakistan, 6 Department of Pediatrics and Child Health, Aga Khan University, Karachi, Pakistan

* mian.haroon@scholar.aku.edu

## Abstract

### Introduction

Medical colleges globally have student organizations that serve to enable students' involvement in research. However, details of their approach and activities are seldom published to serve as learning for student organizations in other settings. The Student Research Forum (SRF), a student organization based at a private medical school in Pakistan aims to facilitate students in acquiring research skills. Following the observation of a downward trajectory of student initiative and interest, SRF leadership restructured the organization and improve its impact. This study describes the development and implementation evaluation of the interventions.

### Methodology

The operational framework was revised using the Theory of Change by the core group. Major interventions included enhanced social media and outreach coordination, research workshops, journal clubs, and mentorship to increase research output, mentorship opportunities, and knowledge of medical research; ultimately improving quality in research. The outcomes generated over the course of the study's duration from July 2019 to September 2021 were analyzed using the process metrics of reach, adoption, and efficacy.

### Results

As a result of the interventions, SRF expanded its reach by conducting a total of 41 events during the duration of the study, facilitated by social media growth on each of SRF's online platforms, with a 300% increase in followers on Facebook, and a nationwide network of 91 student ambassadors. An annual workshop series taught research skills to more than 3800

**Data Availability Statement:** All relevant data are within the paper and its Supporting Information files.

**Funding:** The author(s) received no specific funding for this work.

**Competing interests:** The authors have declared that no competing interests exist.

participants. Students leading their own events, SRF featuring international speakers, and the abstracts submitted to SRF's annual conference, along with the conference's reach of 10,000 students, are seen as improvements in the ToC-informed interventions' adoption. The efficacy of the interventions manifested as the REACH program allocated 56 research projects to vetted applicants.

## Conclusion

The applied interventions have accelerated SRF's progress towards achieving its long-term outcome of increased quality in research as translated by increased research output quantity, mentorship, and knowledge of medical research. Further evaluation is required to assess the success of the ToC. As SRF continues to grow, a continued analysis of the implementation outcomes is imperative to gauge its effectiveness.

## 1. Introduction

Medical colleges the world over have student organizations that serve to facilitate students' involvement in research. Literature indicates that the formation of medical student research societies, coupled with the acknowledgement of perceptions and thoroughly planned interventions, can prove beneficial for all medical students [1, 2]. Initiatives such as research conferences, workshops, student journal clubs, mentorship talks, and providing research opportunities like publications have been shown to increase student interest in research [2, 3].

Globally, student research organizations have made much progress in advancing the research interests of their members. The Harvard College Undergraduate Research Association, an interdisciplinary organization founded in 2007, has connected students with experienced researchers, conducted student research conferences, and publishes a biannual student journal [4]. Such societies exist at a myriad of institutes in higher income countries with the objectives of offering research, mentorship, and skill-learning opportunities [5, 6]. Examples from medical institutes include the Cambridge University Students' Clinical Research Society, Baylor College of Medicine's Student Research Society, and the John B. Graham Medical Student Research Society at the University of North Carolina [7–9]. In the United Kingdom, the National Student Association of Medical Research spreads awareness about academic medicine along with publishing its student journal [10]. Such organizations also exist in medical schools in low- and middle-income countries (LMIC) [2]. In Pakistan alone, medical student research societies can be found in medical institutes throughout the country [2]. Despite this, Pakistan's medical student research output is poor, with only a few institutes generating quality research highlighting the need to explore their approach and enhance its effectiveness [11].

In 1990 the Commission on Health Research for Development stated that strengthening research capacity in LMICs is "one of the most powerful, cost-effective, and sustainable means of advancing health and development" [12]. Despite this, research output in LMICs has been low due to a myriad of reasons—inadequate funding, limited capacity to do research and a lack of a culture that values and sees the importance of research [13]. In addition, a study showed that there is keen interest and involvement in research among final year medical students of Karachi with one of the main factors driving them being their university's encouragement towards research [14]. Another study found that medical students of Karachi felt that research training sessions arranged by nongovernment organizations had a significant role to play in raising research awareness [15]. With this in mind it is imperative to promote and

inoculate a culture of research at an early stage with the resources available so that researchers can be created that, through their work, improve the standing of their country.

The Student Research Forum (SRF) at the Aga Khan University Medical College, a private institution in Karachi, Pakistan is an autonomous organization consisting of medical students, primarily focused on promoting quality research and integrating research opportunities for students between diverse educational specialties. Since its inception in 2004, the mission of SRF has been to make research opportunities more accessible to students within the university. SRF has done this by organizing student research conferences, academic events such as journal clubs and skills workshops, and providing opportunities for students to connect with professionals and mentors. It was the first student-led organization in Pakistan to provide medical student research facilitation and engagement at a national level through conferences and workshops. SRF operated mainly through increasing student and faculty engagement enabling students to be collaborative members of research teams. As the organization expanded, a need was felt to recover from low attendance and ebbing student interest. The SRF core team realized they needed a methodology with an adequate level of operational to facilitate major reforms needed to meet the expanded scope. The team acknowledged the lack of a set protocol to follow. Therefore, from January 2019, SRF sought to develop the Theory of Change (ToC) with subsequent implementation into the operational framework of the organization.

An ascending trend of medical student research engagement in South Asian countries along with SRF's example of using an approach informed by a theory of behaviour change can initiate a cascade of successful student-led endeavors for research [16–18]. This study describes the approach to development, implementation and evaluation of ToC methodology in its first phase to improve processes leading to the overall intended outcome of medical student engagement with research. We also share lessons learned for replication and expansion from the organization's effort to establish its ongoing role to support the practice of research in the institution and in the country. With this paper we aim to examine if our approach can be used to enhance students' engagement with research by the creation of a dedicated student led research society that promotes and imparts skills necessary for research.

## 2. Methods

### 2.1 Setting

Pakistan is home to 122 medical schools, all of which offer "Bachelors in Medicine, Bachelors in Surgery" (MBBS) a 5-year graduate degree. 48 are public institutions and the other 74 private. One of these private institutions is the Aga Khan University (AKU) in Karachi, Pakistan. AKU is an international university primarily based in Pakistan, East Africa, and the United Kingdom. Founded in 1983, AKU's Medical College in Karachi is the first private medical institute of the country and attracts high achievers from all regions within Pakistan, being consistently ranked within the top 10 institutes by the Higher Education Commission (HEC). As part of research training, AKU requires students to pass a compulsory two-week research module during their second year to introduce them to research. However, students are not expected to prepare a manuscript for the purpose of publication in a scientific journal during this module. This is followed by an intensive 8 week module in their fourth year where they are expected to apply their knowledge, formulate an individually designed study protocol, as well as plan and execute a group study with a few of their colleagues.

### 2.2 Study design

This study is a quality improvement project targeting the medical student population of AKU with the goal of enhancing their engagement with research.

The timeline of the study was from July 2019 to September 2021. The study participants consisted of undergraduate students enrolled in medical schools in the MBBS program as well as premedical students enrolled at high schools in Pakistan.

Ethical approval was sought from the Ethics Review Committee at the Aga Khan University and exemption was received (approval number 2022-7137-20351). The individuals named in this paper, along with the information on their involvement in the process, have provided their consent. Participant data is employed only in the form of numbers available from social media platforms and event attendance and is completely anonymized.

## 2.3 Development of ToC

SRF had several objectives which ultimately led to creating a ToC articulating the role of the organization in improving research knowledge and understanding for medical students throughout Pakistan, and ultimately leading to an increase in their academic research outputs. The process for the formation of a theory of change included weekly or biweekly meetings between January to June 2019, with each meeting lasting from 45 minutes to an hour. Meetings would involve updates on a task list e.g. the evaluation and improvement of organization objectives. The fulfillment of these objectives would then be viewed through problems and obstacles for which solutions needed to be created. Progress was recorded as meeting minutes that eventually coalesced to form a primitive ToC. In the following meetings, students in the core group would present a draft of theory of change. Feedback was provided by mentors through questioning about the suggested activities and why this would encourage greater engagement. Principles of the intervention activities were based on incentive, feasibility—free courses, capability of the students, student requirements, and the efficacy of whether these strategies would be effective or not.

**2.3.1 Conceptualization.** SRF employed a novel approach to its strategic planning by formulating a ToC framework and subsequently seeking to evaluate its practical application. De Silva et al. describes the ToC as 'a theory of how and why an initiative works', and that 'which can be empirically tested by measuring indicators for every expected step on the hypothesized causal pathway to impact' [19 2014 p2]. We decided to use the ToC approach as a methodology as it has been effectively utilized to augment research in many instances globally. Universities have incorporated ToC methodology into educational development, evaluation, and research [20]. ToC requires long term goals to be defined and traced backwards in order to identify specific short-term goals needed to be completed in the linear process. While doing so, it is important to document the ToC model showing how and why each goal would be achieved.

At AKU, each student organization is supported by a faculty patron who is responsible for providing mentorship and advice to the organization's leadership. Senior SRF leadership reached out for guidance to improve the condition of the organization. From July 2019 to April 2020, senior SRF leadership and faculty trained in developing intervention and implementation strategies convened over the course of several meetings and formulated the organization's ToC.

**2.3.2 Formation of core group.** The process of developing the ToC involved a group of stakeholders discussing which intervention would provide the most value to the end goal, reflecting on the work, creating transparency, and forming a realistic and complete plan with credible potential for achieving the desired outcome [21]. Several meetings were held by the core group of the SRF for the purpose of planning each event. The core group consisted of a few senior 3rd, 4th and 5th year medical students, all of whom had been involved in research at some stage of their undergraduate medical career (Table 1). All students in the core group had completed the research module and were involved in conducting research at some point

**Table 1. ToC development core group roles and contributions.**

| Name | Role | Domain Expertise | Experience (years) |
|---|---|---|---|
| Adnan A. Khan | Vice President, SRF Medical College | Alumni outreach, event management & strategy aspects of SRF | 5 |
| Ali Aahil Noorali | President, SRF Medical College and School of Nursing<br>Team Lead, Theory of Change project | Leadership and innovation, strategic development and implementation aspects of SRF | 6 |
| Asma Altaf Hussain Merchant | Vice President, SRF Medical College | Inter-societal collaborations, organization and execution of events of SRF | 5 |
| Babar S. Hasan | Faculty Patron SRF | Implementation science, supervision of student organizations | 5 |
| Muneera A. Rasheed | Faculty Global Health | Implementation science, human behavior and engagement | 10+ |

in their undergraduate careers. Participation of these students from the core group was voluntary and based on their interest to improve the SRF.

**2.3.3 Assessment of preconditions.** Understanding the needs of the student community prior to initiating an intervention was necessary to assess its appropriateness. As per the observations of the student members of the SRF core group, diminishing student involvement in research activities at an undergraduate level was juxtaposed with an increasing interest and desire to participate but not having the access to resources to do so. Research experience plays an integral role as a predictor of success for applicants to residency programs in the United States, of which a large proportion of AKU students apply to and therefore is a lucrative asset to possess [22, 23]. Faculty-student mentorship interactions existed perchance, and social media communication was restricted to individual programs (e.g., medicine, nursing, etc.) without a dedicated managing team. There was a lack of research opportunities available for students to translate any interest in research into tangible outputs which was another reason why SRF felt inclined to develop and implement this ToC.

**2.3.4 The model and interventions.** We formulated a causal pathway moving from our desired impact to our preconditions which would ultimately assist in keeping the organization more focused, facilitate communication, and strengthen partnerships and development. The specific intervention activities were designed after discussion and mutual agreement of the core members during the design phase. Being part of the medical college, the core group was very aware of what the reasons were for low engagement of the stakeholders. The main guiding principle was to incentivize engagement through social media and participation of higher leadership. For example, enhanced social media presence of the medical college leadership and their communication was encouraged to enhance faculty engagement and recognition.

Several interventions such as; workshops to impart skills, journal clubs for networking and understanding research, and a program to assign research projects were implemented by SRF with the goals of fostering an environment of research collaboration between senior and junior students, equipping students with basic research skills required to succeed in medical school and providing students the chance to meet with accomplished individuals in their respective fields to create mentorship opportunities (Fig 1).

Furthermore, we developed an evaluative design employing indicators that could effectively measure SRF's future success in maximizing the benefits of making research knowledge more accessible for medical students and those aspiring to join medical schools within the university as well as at the national and international levels [21].

*2.3.4.1 Restructuring SRF.* The first intervention was to restructure and streamline the workings of the SRF. Leadership was split into senior and junior core. Senior core consists of the president from year five of MBBS and three or more vice presidents from year 4, along with an executive council for advisory purposes. Junior core consists of at least two directors per wing

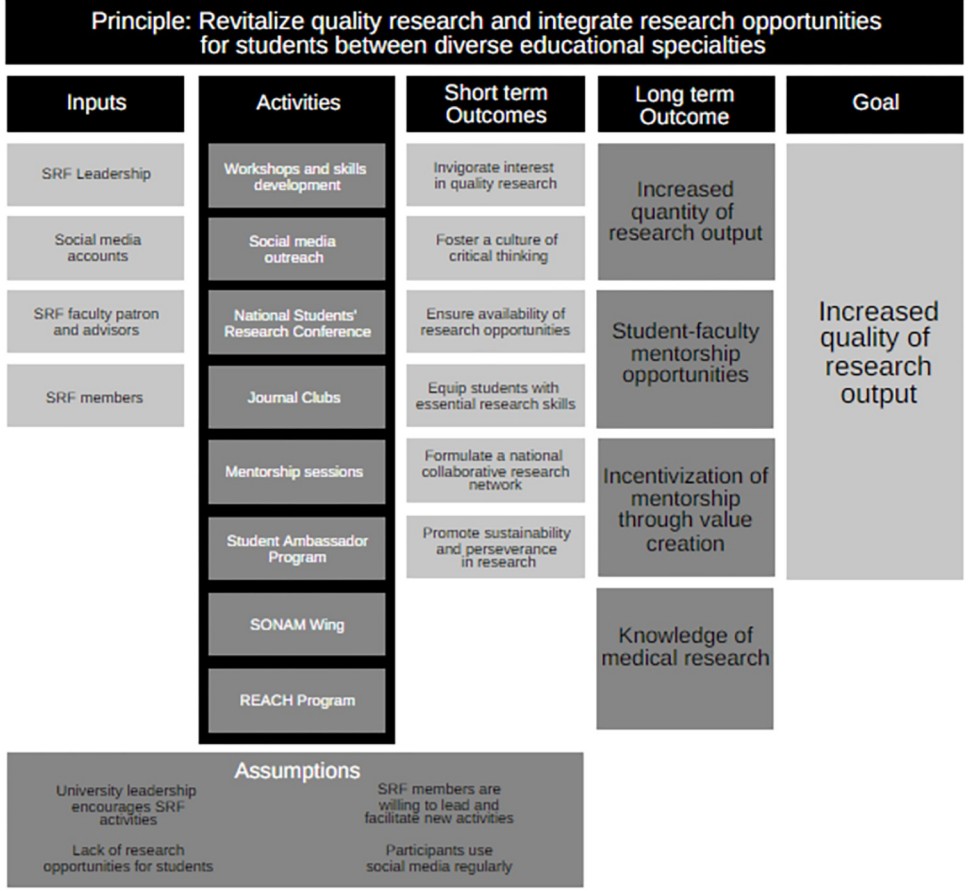

**Fig 1. Logical framework of SRF inputs, activities, and desired outcomes.**

of the organization, with at least three co-directors in each wing. The structure was designed to be inherently flexible at each stratum to change per the current requirement. This is evidenced by the appointment of two presidents concurrently in the academic year 2020–21 to efficiently handle SRF's volume of activities (Fig 2). Members, based on their performance and commitment to the organization, move up the hierarchy to fill in leadership positions and potentially participate in the working core group.

Another intervention was creation of a logo (S1 Fig) to be able to easily incorporate into marketing materials and research posters.

*2.3.4.2 The role of communications and social media.* Social media, specifically, was employed as an essential instrument in connecting with our audience. Platforms such as Facebook, WhatsApp, Instagram, and Twitter were utilized to market information about upcoming events as well as inform our followers about our progress and successful ventures. This included the formulation of a dedicated communications wing for the streamlined generation of marketing and execution of material for all events carried out by SRF.

*2.3.4.3. Student ambassador program.* In early 2020, during the planning stage of the SRF's 8th National Student Research Conference (NSRC), an aggressive strategy designed to develop a student ambassador program aimed to increase outreach and the number of research abstracts received. The student ambassador program was spearheaded by the Administrations Committee of the conference, which was responsible for liaising with institutions and participants to serve as ambassadors for the conference at their institutions and to facilitate

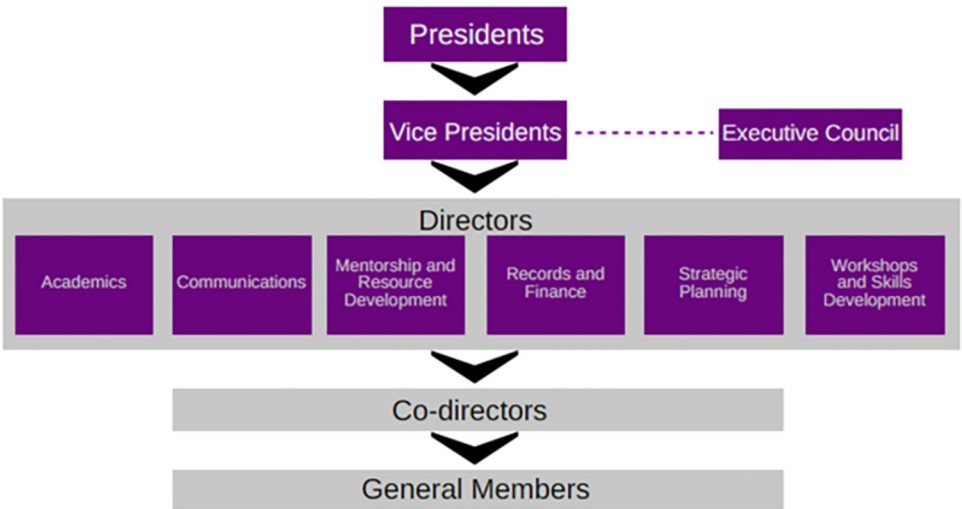

**Fig 2. Administrative structure of Student Research Forum.**

registrations for presentations via the call for abstracts. Volunteers were incentivized by their conference fee being waived as well as a certificate of recognition by SRF and AKU. Conference ambassadors were responsible for information dissemination within their respective institutes while maintaining coordination with the Administrations Committee via email and WhatsApp groups. WhatsApp groups were created within one week of disseminating emails and included both SRF leadership and the recruited ambassadors, allowing quicker communication when compared to conventional email correspondence.

*2.3.4.4 SONAM wing.* As one of its objectives to share its expertise and skills with students of other disciplines, as part of its expansion in accordance with the ToC, SRF formulated a new wing within AKU's School of Nursing and Midwifery (SONAM). The wing started in July 2019 as a semi-autonomous entity under the observation and consultation of SRF senior leadership. SRF SONAM operates under the same framework as that of the parent organization, organizing workshops, research mentorship sessions, and journal clubs to disseminate research related skills and information. In 2021, as the SONAM wing has grown in membership and subsequent activity, it has become a wholly autonomous entity of SRF.

*2.3.4.5 REACH program.* REACH (Research Evaluation and Allocation Committee) was launched by the Resource Development and Mentorship wing in June 2020 with the goal of connecting medical students in search of research opportunities with faculty who wished to expedite their ongoing research projects. The selection process relied on an objective scoring algorithm which accounted for the proficiency of students in skills required by faculty including, but not limited to, conduction of literature reviews, citation management, protocol writing, data collection, statistical/data analysis, and manuscript writing.

Research opportunities were publicized via an online notice and interested students filled out an application requiring self-assessed objective scores of their interest, experience, and proficiency. Completed applications were assessed individually by members of the REACH team and the applicants who scored the highest were shortlisted; the final selection required the evaluation of the CVs of these applicants.

## 2.4 Standard operating procedures

SRF had a standard framework in place that students could follow whenever they had to organize any event, such as a JC or a workshop. Each event was planned months in advance:

relevant faculty members were approached, and the event was marketed in time for students to sign up. Each event had a designated lead who had the responsibility of mobilizing his or her team to complete tasks in time. That team was also responsible for approaching faculty members, holding multiple meetings to brief them on the objectives of the event and target audience, and brainstorming ideas to make the session more interactive. Students were informed of the event through SRF's Facebook, Twitter, and Instagram pages as well as by posting on WhatsApp groups of each class. Students signed up via Google forms and a first come, first served principle allowed us to select our final participants.

## 2.5. Data acquisition and analysis

The Student Research Forum team, based on the framework delineated by Proctor et. al, formulated the definitions of the implementation outcomes utilized in the implementation of the ToC [14]. The data is presented for the first 30 months of implementation (March 2019 to September 2021) and hence the intermediate outcomes of the ToC. Each outcome is measured by its quantitative and qualitative indicators, as listed in Table 2. Per the team's own requirements, implementation outcomes were either adopted from existing literature or defined as a novel and tailored interpretation. The outcomes were decided by the core group during the design based on principles of feasibility of collecting data for the students who had to manage it along with their ongoing classes. Interventions and the implementation outcomes employed in the process are represented in Table 3.

Data acquisition was designated to authors RA and AS. Social media data was collected and downloaded from Facebook, Instagram, and Twitter. Data from Facebook was available through SRF's Meta Business Suite, whereas SRF Twitter data was downloaded through Twitter Analytics. Instagram data had to be manually and individually recorded due to the unavailability of premade profile analysis. Permission for data collection was given by SRF organizational leadership at the time the data was accessed. Data was analyzed using SPSS.

*Reach* is defined as the penetration of the SRF's marketed material in its target population. It is viewed as the initial interaction over the course of the organization's growth with its social

**Table 2. Implementation outcomes and their indicators.**

| Pillar (implementation outcomes) | Quantitative indicators | Qualitative indicators |
|---|---|---|
| Reach | # of members on media platforms or ambassador program | Profiles of members |
| | # of views of social media posts | |
| | % active members | |
| | # of posts uploaded | |
| | # number of reactions | |
| | # of colleges that participated in the conference | |
| Adoption | # of faculty/students who expressed intent to collaborate with students on social media or post workshops | Faculty profile |
| | # of faculty who led workshops and journal club | # of international faculty speaking at SRF events |
| | # of faculty who took on students for research projects | |
| | # of students applying for REACH program | |
| | # of abstracts received for the NSRC | |
| | # of student led research events e.g. workshop sessions | |
| Efficacy | # of papers published by participants through SRF's programs (REACH and JCs) | Medical and research specialties and discussion themes represented in SRF events |
| | # of abstracts/projects presented by medical students at conferences | |

**Table 3. Implementation specifications of the intervention strategies.**

| Strategy | Actor | Action(s) | Target of the action(s) | Temporality | Dose | Implementation outcome(s) affected | Rationale |
|---|---|---|---|---|---|---|---|
| National Students' Research Conference (NSRC) | SRF in its entirety | Create a scientific forum for the exhibition and discussion of student-helmed research. | Medical students and allied health sciences students in Pakistan. Knowledge about research skills, presenting research, and networking with experts. | Preparation begins months in advance with weekly organizational wing meetings. | 3-day conference with daily sessions accumulating to a total of 18 hours. | Efficacy, adoption | Undergraduate research conferences by and for students provide a transformative experience and increase students' confidence [15]. |
| Student Ambassadors Program (SAP) | NSRC Administrations Committee, Communications Wing | Ambassadors disseminate information about the NSRC and National Research Workshop series and facilitate registration for both events. | Medical students belonging to the institute of the ambassador. | Dissemination of information begin 2 months prior to the NSRC and 2 weeks prior to the National Research Workshop series | 60-minute meeting prior to NSRC, communication via email when required e.g during National Research Workshop series 2020. | Reach | SAPs on a volunteer basis allow organizations to establish and maintain communication with students [24]. |
| Social media | Communications Wing | Promote upcoming events and publicize achievements via Facebook, Twitter, Instagram, and WhatsApp. | Followers of the SRF Facebook, Twitter and Instagram pages. Medical students of every class at AKU. Provide regular organization updates and attract potential participants for future activities. | Events are announced on social media at least 2 days prior to the event. A social media post highlighting the successful execution of the event is put up within the next 2 days. | Utilized per the frequency of events happening year-round. Estimated dose of 3 hours per week. | Reach | Social media provides a free and easy-to-use communication medium to reach interested individuals across the country. Social media accounts become advertising sites for an organization's activities and serve to exponentiate outreach. |
| Journal clubs | Academic Wing, faculty from AKU and outside institutions | Discuss and critique impactful studies. Create research opportunities for students. | Medical students of every class at AKU and at other institutes in Pakistan. | Bimonthly meetings by the academic wing to organize journal clubs and liaise with guest speakers. | Journal clubs are held throughout the year. | Efficacy | Journal clubs improve reading and critical appraisal skills, along with increasing knowledge and confidence. [25] Direct contact with experts provides research opportunities. |
| Workshops | Workshops and Skills Development Wing | Disseminate research skills. | Medical and high school students in Pakistan. | Bimonthly meetings by the workshops and skills development wing to organize sessions and liaise with guest speakers. | Workshops are conducted throughout the year. 3-day National Research Workshops Series held annually. | Efficacy | Skill based workshops accessible to participants across the country expand students' skill sets and equip them with the knowledge and tools to conduct their own studies. |

(*Continued*)

**Table 3.** (Continued)

| Strategy | Actor | Action(s) | Target of the action(s) | Temporality | Dose | Implementation outcome(s) affected | Rationale |
|---|---|---|---|---|---|---|---|
| REACH Program | Resource Development and Mentorship wing | Connect students to faculty with research opportunities. | Medical students at AKU. | Meetings should be held weekly to identify needs and develop the algorithm. | Weekly 1-hour meetings to develop the research allocating algorithm. Upon implementation, the REACH program is operational throughout the year. | Adoption, Efficacy | The lack of communication between students and faculty prevents students from gaining the advantage of research experience and mentorship. REACH bridges this gap by identifying research openings objectively evaluating student applications to allocate opportunities. |

media followers. Indicators include the number of followers on social media platforms, marketing penetration via the student ambassador program, the number of views of social media posts, the percentage of active members, the number of posts uploaded to social media platforms, the number of reactions to social media activity, and the number of colleges participating in the NSRC. Qualitatively, *reach* is indicated by the profiles of followers as portrayed by verified status and social media influence.

*Adoption* is defined as the uptake of the SRF's research facilitating activities in individuals with whom there is direct interaction, including attendees, facilitators, and organizing, as well as any other parties that may be involved. This is an interpretation of Proctor et al's adoption implementation outcome and focuses on the translation of SRF's activities into the research opportunities that have been generated as a result of said activities [14]. Indicators of this implementation outcome include the number of faculty or students who expressed intent to communicate, collaborate, or teach participants on social media or through research workshops, the number of faculty who facilitated workshops and journal clubs, the number of faculty who took on students for research projects, the number of students who have applied to the REACH program, and the number of abstracts received for the NSRC. The presence of international faculty speaking at SRF events and faculty profile have also been listed as qualitative indicators to see the development of SRF's network and availability of research access to SRF participants from a global perspective.

*Efficacy* is the most precise of the implementation outcomes and aims to look at the transformation of *reach* and *adoption* into concrete manifestations of the research opportunities generated by the SRF as well as the diversity of research outputs produced by the participants. These include the number of research articles published in journals as well as the number of projects and/or articles presented at research conferences. The variety of themes and topics represented in SRF's events are a qualitative indicator of *efficacy* and aim to portray the extent of SRF's involvement in communicating a heterogeneous milieu of research avenues.

## 3. Results

In this section the details of each intervention and what they entailed have been discussed. For clarity, all the events, with their names, have been stratified in Table 4 by year and then

**Table 4. SRF events and activities divided by year.**

| 2019 | 2020 | 2021 |
|---|---|---|
| STATA 101[1] | The Roundtable A Journal Club Series: Episode 7[1] | The Roundtable: A Journal Club Series Episode 12[1] |
| Meet the Matched- 2019[1] | The Research Games[1] | Inventing the World's First Portable MRI: The Role of Research in Industry[2] |
| The Roundtable A Journal Club Series: Episode 2[1] | Meet the Matched 2020[1] | Meet the Matched 2021[1] |
| Case Report Writing 101[1] | The Roundtable A Journal Club Series: Episode 8[1] | Live Demonstration: Hyperfine MRI Machine[1] |
| The Roundtable A Journal Club Series: Episode 3[1] | The Roundtable A Journal Club Series: Episode 9[1] | Crossing Borders: An Insight to the UK Training Experience[2] |
| The Roundtable A Journal Club Series: Episode 4[1] | What happened to my paycheck?[1] | From a Jester to An-King: Mastering the Basics of Anki[1] |
| Women in Research[1] | The Roundtable A Journal Club Series: Episode 10[1] | Research & SRF: An Introduction[1] |
| 1st Chai and Chat[1] | A Beginners Guide to Research[2] | Personal Financial Management During Residency in the US[1] |
| The Roundtable A Journal Club Series: Episode 5[1] | Research 101[2] | Med School and Beyond: Insta Live with SRF[2] |
| 2nd Chai and Chat[1] | Anki 101: A Beginner's Guide on How Not to Forget Stuff[1] | The Roundtable: A Journal Club Series Episode 13[1] |
| The New Era of Research: Data Science in Health[1] | The Roundtable A Journal Club Series: Episode 11[1] | From Squires to Knights[1] |
| The Roundtable A Journal Club Series: Episode 6[1] | SRF Alumni Series Episode 1: Meet the Experts[1] | The Roundtable: A Journal Club Series Episode 14[1] |
| | SRF Alumni Series Episode 2[1] | Mentorship From Seniors[2] |
| | SRF Alumni Series Episode 3[1] | Stats with SRF[1] |
| | | Meta Analysis: Everything You Need To Know[2] |

1 = Open to home institute (AKU) participants only

2 = Open to all participants irrespective of institute

presented in a chronological order. The implementation evaluation of the results is based on the metrics provided in Table 2.

Additionally, events have been tabulated in S1 Table with additional details to further breakdown and display the name of the event, the theme, the number of participants, the organizing wing, along with an outcomes and description column.

## 3.1 Journal clubs

A total of 13 journal clubs (JC) were held during the period of this study. JCs featured guest speakers from a variety of medical disciplines, including cardiothoracic surgery, general surgery, trauma surgery, pulmonology, cardiology, neurology, infectious disease, gastroenterology, and psychiatry. Journal clubs resulted in opportunities for students to work on research papers, with many of these papers being published. A notable example is that of the second SRF JC on cardiothoracic surgery with Dr. Saulat Fatimi, which resulted in 20 students publishing case reports and 4 students publishing full length articles. The provision of research opportunities resulted in an aura of intrigue and enthusiasm and student interest in SRF JCs effectively increased by word-of-mouth within the student community. In the following editions of JCs, students were able to garner research opportunities and develop mentoring relationships.

Apart from learning about advances in medical subspecialties, journal clubs also provided participants opportunities in which they learned how to critique and analyze research articles. This is indicated by the sessions held with Dr. Ayeesha Kamal and Dr. Faisal Mahmood, with Dr. Mahmood discussing the arrival of the COVID-19 pandemic resulting in a high volume of research being published in a relatively short period of time and its veracity and reliability.

As sessions went on and the team's expertise in organizing such events increased, SRF JCs began to feature experts from top notch institutes from other parts of the globe. Dr. Mahbob Alam, a cardiologist at Baylor University, Dr. Hasan Alam, chair of surgery at Northwestern University, and Dr. Zain Sobani, a gastroenterologist at Augusta University, were among the international faculty represented. A milieu of distinguished guest speakers from both within Pakistan and from other countries, along with the research and networking opportunities made available, further established JCs as one of SRF's cornerstone events.

### 3.2 Workshops

Seven workshop-based activities took place during the study's timeframe. Standalone workshop events range from teaching the application of analytical software such as SPSS and STATA to imbibing the art of penning research articles and carrying out meta-analyses. Longer, more meticulously planned series of workshops were also conducted. The first of these was called 'A Beginners Guide to Research', which consisted of 4 online workshops focusing on an introduction to research & medical ethics, conducting a literature search, basics of abstract and manuscript writing and referencing. This series was conducted by both faculty members and senior medical students. More than 3800 medical students from all over Pakistan signed up to attend these workshops [26]. Following this, the 'Research 101' workshop series catered to more than 1200 pre-medical and high school students, inculcating a more simplified version of the lessons taught in the 'A Beginners Guide to Research' workshops S1 Table. The workshops and skills development wing replicated successfully executed workshops to teach new cohorts of participants, along with designing novel sessions, many of which happened after the end of this study's duration.

### 3.3. REACH program

During the course of the study, the REACH program had assigned 36 SRF members and 20 non-SRF students to different projects with various faculty members. Most of these projects are currently in the process of completion or have been submitted for journal review. The program is being upscaled in collaboration with other student organizations under the guidance of the medical college administration. Students with respectable academic standing and an interest in research will be able to apply to work for research projects, regardless of their affiliation to SRF.

### 3.4. Student ambassador program

The student ambassador program resulted in 91 ambassadors being selected from a pool of more than 400 applicants. The ambassadors represented 46 health sciences institutions in 66 programs across the country, which are listed in S2 Table. The majority of ambassadors were enrolled in medical and health-related programs such as MBBS and BScN, however, programs such as bachelor's in business administration (BBA), bachelors of science in bioinformatics, science in biotechnology, and of science in microbiology were also represented. In purview of the postponement of the NSRC and SRF's new online approach, the robust student ambassador program was put to a novel use. Ambassadors were now contacted to increase outreach for the National Research Workshop series, with separate streams for undergraduate and high

school students. More than 3800 participants attended the sessions from 123 institutes across the country.

### 3.5 National student research conference

Despite being one of the flagship events conducted by the organization, SRF was not able to hold a research conference in 2020 because of the limitations presented by the COVID-19 pandemic. SRF was able to disseminate the call for abstracts for the conference, which resulted in more than 10,000 students being reached. Conference organizers received more than 140 abstract submissions within the first 3 days of the announcement, following which the postponement notification was uploaded to our social media accounts. The planning phase also yielded confirmations of attendance by keynote speakers from health sciences institutes in the United States.

### 3.6 Mentorship sessions

During the course of the study duration, SRF conducted 16 mentorship related events (listed in Table 4 and S1 Table). These sessions effectively develop and expand SRF's network of accomplished experts of their respective institutes and fields, including neurology, cardiology, endocrinology, psychiatry, community health sciences, biomedical sciences, general surgery, pediatrics, healthcare management, and business administration. The majority of these activities were helmed by the resource development and mentorship wing. However, one of the organization's flagship events, 'Meet the Matched', an annual session that showcases the journeys of AKU Alumni into residency programs in the United States and the impact of research in their application process, requires the cooperative effort of all wings. SRF members temporarily put their wing-related responsibilities on hold and direct their focus on planning the year's 'Meet the Matched' event. In April 2019, the session attracted 300 attendees. With the advent of the pandemic and a shift towards remote modalities of communication, a venue attendee limit was effectively removed, and in the 2020 edition of the event, more than 300 individuals were able to participate, including those not present in Karachi, Pakistan. This was followed by 2021's 'Meet the Matched' event, which was held online as well, and catered to more than 300 participants as well.

### 3.7 Social media and outreach

Social media metrics were analyzed over the course of the study duration. Our online presence on our three social media accounts, namely Facebook, Instagram, and Twitter were evaluated. Data made available by the respective platforms is represented in Table 5.

a.  SRF's Facebook presence is the largest out of its social media. At the end of the study period (September 2021), SRF amassed more than 6,957 page likes, along with more than 7,200 page followers (Fig 3).

b.  Instagram possessed the second highest following of SRF's social media accounts at 2702 at the end of the study duration. Over the course of the study, 285 posts were published with a total number of likes of 13634.

c.  Twitter analytics were analyzed from April 2019 to November 2021 due to limited availability from the platform (Fig 4). The number of followers at the end of the study duration amounts to 1781.

**Table 5. Engagement trends of the social media accounts over the study period.**

| | Facebook | Instagram | Twitter |
| --- | --- | --- | --- |
| | **Timeline (1/1/19–30/9/21)** | **Timeline (5/4/19–30/9/21)** | **Timeline (5/4/19–30/9/21)** |
| Total N (%) of active members | 6957 Page likes (as of **30/9/21**)<br>• Gender<br>○ 40.8% men<br>○ 59.2% women<br>• Age<br>○ 61.1% between ages 18–24<br>○ 29.8% between ages 25–34<br>○ 6.3% between ages 35–44<br>○ 2.8% between ages 45+<br>• Location<br>○ Pakistan 93.3%<br>○ United States 2.3%<br>○ Canada 0.6%<br>○ United Kingdom 0.6%<br>○ Afghanistan 0.4%<br>○ United Arab Emirates 0.4%<br>○ Kuwait 0.3%<br>○ Saudi Arabia 0.2%<br>○ Australia 0.1%<br>○ Syria 0.1%<br>**Facebook Page Reach[1]**<br>126,680 | 2702 followers (as of **30/9/21**)<br>• Gender<br>○ 36.7% men<br>○ 63.3% women<br>• Age<br>○ 1.6% between ages 13–17<br>○ 70.9% between ages 18–24<br>○ 23.4% between ages 25–34<br>○ 2.0% between ages 35–44<br>○ 1.8% between ages 45+<br>• Location<br>○ Pakistan 93.1%<br>○ United States 4.1%<br>○ Canada 0.3%<br>○ United Kingdom 0.5%<br>○ United Arab Emirates 0.3%<br>Engagement: 681.60 | 1781 followers (as of **30/9/21**)<br>Total engagements: 42810 (mean: 121.62<br>And range 0–1547) with an engagement rate of 7.17 |
| N posts and reach | 227 posts with a<br>reach of 565012<br>(mean: 2489.04<br>and range 0–25382) | 285 posts with a<br>reach of 128264<br>(mean: 450.05<br>and range 0–2827) | 352 posts with total 629324 impressions (mean: 1700<br>And range 23–17161) |
| Total reactions | 19415 Likes and reactions<br>1199 shares | 13634 Likes<br>775 shares | 4165 likes<br>1163 retweets |
| Mean reactions/post Range | Likes and reactions<br>• Mean: 85.53<br>• Range: 1–1193<br>Shares<br>• Mean: 5.28<br>• Range: 0–111 | Likes<br>• Mean: 47.84<br>• Range: 0–564<br>Shares<br>• Mean: 2.72<br>• Range: 0–111 | Likes<br>• Mean: 11.83<br>• Range: 0–132<br>Retweets<br>• Mean: 3.30<br>• Range: 0–53 |
| Total comments | 2378 | 238 | 294 replies |
| Mean comments/post Range | Comments<br>• Mean: 10.48<br>• Range: 0–237 | Comments<br>• Mean: 0.84<br>• Range: 0–24 | Replies<br>• Mean: 0.84<br>• Range: 0–8 |

[1]number of unique people who saw any content from Page or about Page

To continue fulfilling its purpose in accordance with the ToC, SRF shifted its focus to planning online events during the Covid-19 pandemic. Communications platforms such as Zoom and Microsoft Teams were utilized to conduct these events and fill the gap in co-curricular activities left by the pandemic within the AKU student community, allowing SRF to become an early adopter within AKU student organizations to harness the advantages of remote communication in an otherwise bleak and isolated period.

## 4. Discussion

The objective of this paper is to explore our approach can be used to inform interventions that enhance students' engagement with research through a dedicated student led research society that promotes and imparts skills necessary for research. Analysis of the results using the implementation outcomes defined above; reach, adoption and efficacy have shown that the

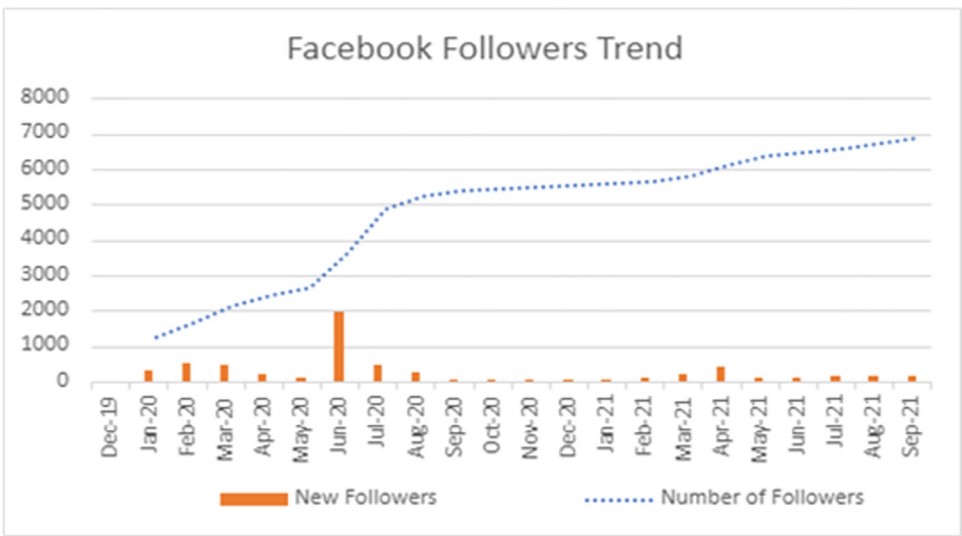

**Fig 3. Trend of total Facebook followers and new followers per month during the study duration.**

interventions have largely been successful in achieving most of the intermediate outcomes, namely creating mentor-mentee connections (through JCs), orienting other national institutions to a culture of academic research to by creating student ambassadors and creating value around research using social media. We discuss each outcome one by one below.

## 4.1 Reach

Facebook was our largest social media presence with 6,900 likes and 7,200 followers, which was a drastic increase of approximately 300% from before implementation of the ToC. Instagram and Twitter were other social media websites that SRF used and were 2nd and 3rd largest

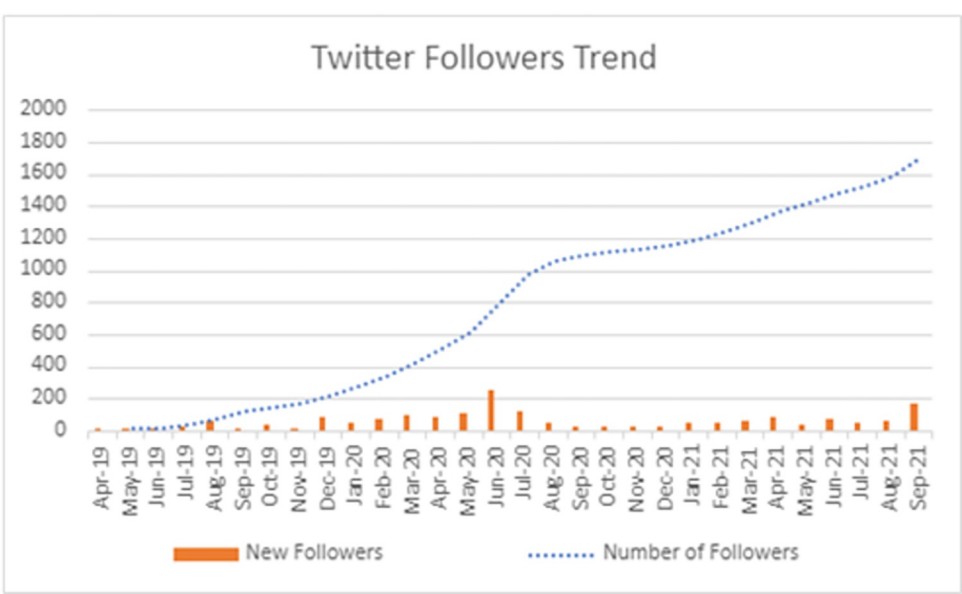

**Fig 4. Trend of total Twitter followers and new followers per month during the study duration.**

respectively. For Facebook and Instagram, the vast majority of followers were located in Pakistan (93.3% and 93.1% respectively), mainly because the university is located in Pakistan and most marketing targeted students located in this country. The accounts managed to obtain a slight follower presence in other countries as well such as the United States, Canada, the UK, and the Middle East as some of our alumni are based there and the international student population are predominantly from these countries. This improvement in the social media presences can be attributed to several factors. A dedicated communication wing was formed that was solely focused on the generation of marketing material for all events carried out by SRF. There was an increase in the quality of posts and marketing material such as posters and write ups. Professional editing software such as Photoshop and Adobe Premier Pro were now used to create the materials. Students well versed in editing helped create templates for posters and marketing material. The posters were sent to the society leaders to be vetted and mistakes ironed out before posting on social media. Finally, for our highly anticipated events, such as Meet the Matched, we created videos to build interest and market the event. There was the added advantage of Pakistan being in one time zone (GMT+5). Posts during non-work hours and during the weekend have higher engagement [27]. Realizing this, we posted at suitable times to obtain maximum engagement from the majority of our followers which are based primarily in Pakistan.

The increased reach obtained by having a greater social media presence allowed an increasing number of students and faculty from other institutes to be exposed to the posts, which led to more signing up and being provided the opportunity to learn essential skills required for research. A benefit of an online modality that was adopted, marketed, and hosted during the COVID pandemic on social media was that it was not limited by money or different geographical locations. This meant that more students could access the content that we were providing if they had a device that connected to the internet and an internet connection. The COVID pandemic may have facilitated participation through means of a flexible schedule, allowing more participants to join. An increase in laptop acquisition during the pandemic could have also contributed to increased participation [28]. In addition, social media created value around research by increasing student engagement which is a significant step to improving quality of student research–by creating and developing interest in research first.

The student ambassador program was also a method utilized to increase the reach that SRF had. It allowed us to gain a direct link with 46 other health science institutions in Pakistan permitting marketing directly to these universities. Through these ambassadors, more students came to know about SRF, which might not have been possible through online marketing only. People are more likely to trust a product, if informed about it from someone they know personally, through word of mouth [29]. These ambassadors were contacted to promote NSRC 2020 and then later to market the National Workshop Series. Both events had a large volume of signups. More than 10,000 students were reached when announcing NSRC before postponing it and for the National Workshop Series greater than 3800 participants attended from 123 institutes. This amount of engagement of interest in these events may not have been possible without the network of student representatives SRF had already established. This intervention fostered relationships between AKU and other universities that these ambassadors were a part of. Additionally, those students who were selected as ambassadors benefitted as well as they were given a chance to develop their leadership and communication skills by taking an active role in representing their university and at the same time forging new connections and networks.

The new logo that was developed served as a banner head that SRF could fall under, made the society identifiable with just an image, and allowed for production of more professional marketing materials.

## 4.2 Adoption

Research conferences are an important experience for students, especially at the undergraduate level to step outside their comfort zone, develop their interpersonal skills and increase their confidence [15]. They are important drivers in terms of publications as well as students interact with other fellow researchers with the same interests, network and collaborate, leading to more publications and dissemination of knowledge [30]. Although the NSRC had to be postponed because of the COVID 19 pandemic, in the 3 days since its announcement, more than 140 abstracts were submitted. After the postponement, SRF received numerous emails asking when the conference would resume and abstract submission be reinstated, indicating continued interest in the conference.

Students in general are interested in delving into research and faculty are always looking for eager students to help them in their projects. REACH aimed to become the bridge between the two and provide access of both parties to each other. By finding willing faculty and attaching students that want to learn, valuable skills essential to research are retaught and a mentor-mentee relationship is formed. As the selection process is skill based, students can work up from easier to more skillful tasks and learn what skill set they are deficient in, making them better equipped in the long run.

An interesting finding of this study is that after the introduction of the ToC-informed interventions, there has been an increase in student-led events with students themselves as the ones who are talking to or teaching the audience. These sessions were mainly either tutorials on how to use a certain programme, such as Anki (a spaced repetition flashcard programme) or talks on the basic skills needed in research. This indicates that students were confident to take initiative and in their abilities and research skills that they were willing to teach others, potentially increasing the research capabilities of the whole student body. These sessions were instrumental in inspiring juniors as it showed them that fellow students can and have reached this level of mastery.

## 4.3 Efficacy

Journal Clubs were conducted with the aim to enlighten students with the latest strides being made in that particular discipline. Moreover, it gave students research opportunities. For example, attending the 2nd cardiothoracic surgery JC can be directly attributed to 20 students publishing case reports and 4 students publishing full length articles. For many participants, especially the 1st years that attended, this was their first paper published. In addition, most of the projects being case reports is significant as they are relatively simpler projects and are essential stepping stones for newly budding researchers to develop their skills. JCs were also impactful in other ways, such as allowing direct sharing of ideas between students and faculty. Additionally, students were able to see a polished and finalized study with a large impact, learning directly from the author themselves the process that they had to undergo to reach this stage. Multiple JCs were held with different specialties, enabling students to network with faculty in the field of their interest and forge a connection.

Although it was not possible to qualitatively measure if any of the workshops or the mentorship SRF facilitated directly led to any publications, one may assume that by teaching a person the required skills to do research, will eventually lead them to contributing to a project. Holding regular workshops, both at the university level and nationally, equipping students with the required skill set to conduct their own standard research and giving them confidence in their abilities all promoted production of quality research and invigorated students' interest in it.

The rapid growth of SRF and the scope of the society's events can be largely attributed to the creation of separate wings that were delegated distinct responsibilities. The reason why this

intervention was successful was because it allowed the SRF to plan or execute multiple events concomitantly without affecting efficiency of the society on a whole. For example, the Academic Wing would be planning their next JC, while the Workshop Wing would be working on their next workshop, and this would happen without either wing impeding the other. This allowed us to have multiple, logistically intensive events in quick succession. In addition, instead of having multiple people involved with the entire workings of the society, having specific directors specialized in the workings of their respective wings ensured that the society ran more effectively. The specific directors had a tailored skillset required for the working of their wing. This eased the burden on the leadership, such as the presidents and vice-presidents, gave more structure to the events going on and allowed easier delegation of tasks. This structure proved to be extremely advantageous and flexible as demonstrated during the COVID-19 pandemic, where the society had to shift from in person and physical events to an online modality. Eventually, the wings soon reached a high level of expertise that they were able to expand their efforts to a nationwide level. Details of each wing and their role in the SRF has been further elaborated on in the S1 File. Moreover, being able to see the effect of their interventions on interest of beneficiaries and faculty helped to maintain their engagement. Collecting insights from ongoing implementation and acting on is vital for success of behavior change interventions.

Although our study evaluates the success of our interventions in achieving the objectives of SRF using a ToC guided framework, we believe the study also provided students an opportunity to design their activities not with just an outcome in mind but also with the understanding of why this would change organizational behavior. This skillset we believe is a core element of leadership abilities—how to influence human behavior.

## 4.4 Limitations

There are some limitations in the ToC that was implemented. The most glaring is that there were no measures in place to assess the effectiveness of teaching activities conducted during JCs and workshops. Hence, we are unable to conclusively determine whether the strategy was effective or not. The majority of sessions had no feedback, so direct input from participants was rarely collected. We observed that it was also not possible to find out what exactly participants are able to publish after attending sessions that SRF arranged, unless the participants directly reach out, as there was no concrete follow up system. Any other characteristics of the participants that benefitted from the interventions, such as their year of study, program of study, and university affiliation, amongst many others, were also not recorded.

The outcomes that we were able to measure were intermediate outcomes such as; student mentorship opportunities, increased knowledge/awareness of research concepts (through workshops, conferences, JCs), creating value around research. We could not measure the short- term outcome for this phase which was student research outputs due to a lack of resources. That being said, it is still being targeted in an ongoing process and the metric that will be used to measure this final outcome is yet to be defined.

Another point of note is that exponential growth is never sustainable, it has to plateau. The rapid increase in interest in SRF, the rise in follower count and the surge of engagement will eventually decline. With most of our engagement being online, the fact that Pakistan is a LMIC where people have limited access to the internet, electricity and electronic devices is a fact that cannot be ignored. Finally, it is imperative to acknowledge that the research culture at Aga Khan University, an elite private institution, has provided major impetus for SRF's activities in the form of resources and opportunities that may not be available elsewhere in institutes in Pakistan and other low- and middle-income countries (LMICs), effectively constraining the

generalizability of our study's findings. Having said that, creating a ToC considering the contextual needs can help achieve the outcome of increasing student research outputs.

## 4.5 Future

Based on its current trajectory, it is vital for SRF to maintain a thorough understanding of what role the organization sees itself fulfilling in the future. An inclination towards expansion of the organization's activities and scope within students' research requires a well-delineated strategy. An empirical evaluation of the role of ToC while going forward, supplemented by internal restructuring where necessary in the form of internal auditing and the maintenance of records may facilitate the development of new ToC-based ideas and interventions. Examples of such interventions include the creation of a research journal to feature students' research achievements, collaboration with other research-based entities in the form of sharing expertise and conducting events, and the formation of an international alumni network to support students' research ambitions and provide relatable mentorship based on a shared understanding of SRF's mandate. Furthermore, to ensure the enhancement of quality of research output, the Dean's Office should be taken on board. A medical student's main priority is the completion of their degree, which requires learning a huge expanse of medical knowledge while at the same time performing their clinical responsibilities. Enhancing quality of research output would be a huge undertaking, that would be out of the scope for an undergraduate medical student. Hence, this responsibility should be led with the Dean's Office and university administration taking an active role in this with collaboration with SRF.

## 5. Conclusion

We have found that the applied interventions have played an effective role in the improvement and upscaling of a medical student-led research organization at a private medical school. A well-planned and vigorous approach to enhance research engagement as interpreted by the implementation outcomes of reach, adoption, and efficacy has proved to be fruitful. These interventions are effective, but partly due to the fact that these were informed by the theory of change. The ToC was designed by a multidisciplinary team and framed by a behavioral scientist with much deliberation on *why* the proposed intervention would work. The flexibility of the interventions has allowed the organization to achieve its goals within the context of unforeseen adversity such as the COVID-19 pandemic. Replication of these interventions, informed by a well-designed ToC model, in other student settings in LMICs can be feasible.

## Supporting information

**S1 Fig. Development of SRF logo from 2004 to present date.**
(TIF)

**S1 Table. Events organized by SRF during the study duration.**
(PDF)

**S2 Table. List of student ambassador program ambassadors' institutes.**
(PDF)

**S1 File. Description of SRF wings.**
(PDF)

## Author Contributions

**Conceptualization:** Mian Arsam Haroon, Ali Aahil Noorali, Asma Altaf Merchant, Adnan Ali Khan, Sana Gul Baloch, Arsal Tharwani, Saulat H. Fatimi, Zainab Samad, Babar S. Hasan, Muneera A. Rasheed.

**Data curation:** Mian Arsam Haroon, Ali Aahil Noorali, Abdullah Saeed Khan, Muzamil Hamid Hussain, Rohan Advani, Ashmal Sami, Asma Altaf Merchant, Adnan Ali Khan, Sana Gul Baloch, Arsal Tharwani.

**Formal analysis:** Mian Arsam Haroon, Abdullah Saeed Khan, Muzamil Hamid Hussain, Rohan Advani, Ashmal Sami.

**Investigation:** Mian Arsam Haroon, Ali Aahil Noorali, Abdullah Saeed Khan, Muzamil Hamid Hussain, Rohan Advani, Ashmal Sami, Adnan Ali Khan.

**Methodology:** Mian Arsam Haroon, Ali Aahil Noorali, Abdullah Saeed Khan, Muzamil Hamid Hussain, Asma Altaf Merchant, Adnan Ali Khan, Babar S. Hasan, Muneera A. Rasheed.

**Project administration:** Mian Arsam Haroon.

**Resources:** Mian Arsam Haroon, Ali Aahil Noorali, Abdullah Saeed Khan, Muzamil Hamid Hussain, Rohan Advani, Ashmal Sami, Asma Altaf Merchant, Adnan Ali Khan, Sana Gul Baloch, Arsal Tharwani, Saulat H. Fatimi, Zainab Samad, Babar S. Hasan.

**Software:** Mian Arsam Haroon, Rohan Advani, Ashmal Sami, Arsal Tharwani.

**Supervision:** Mian Arsam Haroon, Ali Aahil Noorali, Sana Gul Baloch, Arsal Tharwani, Saulat H. Fatimi, Zainab Samad, Babar S. Hasan, Muneera A. Rasheed.

**Validation:** Mian Arsam Haroon, Ali Aahil Noorali, Abdullah Saeed Khan, Muzamil Hamid Hussain, Ashmal Sami, Asma Altaf Merchant, Adnan Ali Khan, Sana Gul Baloch, Saulat H. Fatimi.

**Visualization:** Mian Arsam Haroon, Abdullah Saeed Khan, Muzamil Hamid Hussain, Rohan Advani, Ashmal Sami.

**Writing – original draft:** Mian Arsam Haroon, Abdullah Saeed Khan, Muzamil Hamid Hussain.

**Writing – review & editing:** Mian Arsam Haroon, Abdullah Saeed Khan, Muzamil Hamid Hussain, Sana Gul Baloch, Arsal Tharwani, Saulat H. Fatimi, Zainab Samad, Babar S. Hasan, Muneera A. Rasheed.

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
