## [Decision Letter · Decision Letter 0]

17 Aug 2022

PONE-D-22-11271Implementing a medical student led theory of change to enhance students’ engagement with research: findings and lessons learnedPLOS ONE

Dear Dr. Haroon,

Thank you for submitting your manuscript to PLOS ONE. After careful consideration, we feel that it has merit but does not fully meet PLOS ONE’s publication criteria as it currently stands. Therefore, we invite you to submit a revised version of the manuscript that addresses the points raised during the review process.

Please note that we have only been able to secure a single reviewer to assess your manuscript. We are issuing a decision on your manuscript at this point to prevent further delays in the evaluation of your manuscript. Please be aware that the editor who handles your revised manuscript might find it necessary to invite additional reviewers to assess this work once the revised manuscript is submitted. However, we will aim to proceed on the basis of this single review if possible. The reviewer has identified a number of significant concerns which need to be carefully addressed in your revision. Please pay particular attention to the queries they have raised regarding the rationale and context of the study in the theory of change, as well as the need to include additional detail about the model and data analysis.

We look forward to receiving your revised manuscript.

Kind regards,

Jamie Males

Editorial Office

PLOS ONE

Journal Requirements:

2. Please ensure that you refer to Figure 1-4 in your text as, if accepted, production will need this reference to link the reader to the figure.

3. Please include table 5 as part of your main manuscript and remove the individual files. Please note that supplementary tables (should remain/ be uploaded) as separate "supporting information" files

Reviewers' comments:

Reviewer's Responses to Questions

**Comments to the Author**

1. Is the manuscript technically sound, and do the data support the conclusions?

Reviewer #1: Partly

2. Has the statistical analysis been performed appropriately and rigorously? 

Reviewer #1: I Don't Know

3. Have the authors made all data underlying the findings in their manuscript fully available?

Reviewer #1: Yes

4. Is the manuscript presented in an intelligible fashion and written in standard English?

Reviewer #1: Yes

5. Review Comments to the Author

Reviewer #1: This study aims to describe how implementing a theory of change led to increasing student engagement with research. A theory of change is a description of how an intervention is intended to lead to outcomes for beneficiaries and changes the focus of concern from what is being done, to what is to be achieved. Once a theory of change is in place, it can be used to evaluate the outcomes and establish whether and how an initiative works. If failure of the theory is observed, then questions can be asked about whether that was due to the theory being adopted (i.e. whether the theory is flawed) or whether the interventions need to be changed. The study describes how interventions were put into place through the vehicle of a Student Research Forum to improve student engagement with research and then evaluated on key metrics of reach, adoption and efficacy. The study concludes that the reach of the Student Research Forum increased and efficacy was measured based on the number of research projects resulting from the engagement.

Recommendations

1. Ethics. The authors have indicated that no ethical statement was needed for this study. However, in any research or evaluative study there are ethical implications for how it is carried out, even if no formal ethical approval has been sought. I had a number of concerns relating to ethics in this paper, first and foremost that it is not considered at all in the text. In addition, participants names are used, as well as organisational and role details. In and of itself, this may not be an issue, providing that permission has been sought from participants, and any unintended consequences have been considered. If this is the case, then a paragraph needs to be inserted to explain this process and the resulting ethical decisions. If not, then I would suggest anonymising any data pertaining to individuals. I would recommend referring to the British Educational Research Association (BERA) guidelines in this process as a starting point.

2. The use of theory of change. The title implies that the paper explores the efficacy of the theory of change process. This however does not materialise and, rather, the paper explores the reach and efficacy of the interventions rather than the theory of change as an approach. I would recommend that either the title is changed to reflect this, or that more details are provided about the theory of change process itself. This needs to comprise a more detailed examination of literature concerning theory of change approaches situated in a consideration of alternative approaches that could be used. The paper then needs to explain the limitations of the approach as well as the potential benefits and provide an explanation in the conclusion about any challenges or successes that emerged from the theory of change approach.

3. Interventions. It is not clear to me how the theory of change process led to the implementation of particular interventions. I would recommend considering the evidence for particular interventions in your literature review and then explaining how the adoption of a theory of change approach led to the authors (and others?) deciding these were appropriate interventions.

4. The model. I recommend giving a description of how the theory of change approach was carried out. What kinds of activities were used to develop the theory of change? How was consensus reached? How did the theory of change help in deciding what to do?

5. Data analysis. ‘implementation outcomes were either adopted from existing literature or defined as novel and tailored interpretation’. This needs explaining and more detail given. What literature? How were they decided?

6. Results. These are results of the evaluation based on reach, implementation and efficacy, rather than the results of using a theory of change approach and this should be made clear, and the metrics for measuring reach, adoption and efficacy clearly outlined at the beginning of the section, or in previous section around methods. The methods need to state how this information was collected, what form it was in, how it was analysed and what permissions were sought to use it.

7. I remain unsure of how Figure 1 relates to the theory of change. The outcomes for students are listed as objectives, rather than steps in the theory to enhance student engagement with research. The intermediate outcomes seem more like a list of measurements than expected changes. I would recommend reading and referring to work by Carol Weiss, and also by Mackenzie and Blamey, and then finally by Connell and Klem. Pawson’s work also provides a wider overview of the field.

8. Discussion. The paper does not show that the theory of change has been successful in achieving outcomes as it is claimed. It shows that the interventions have been successful in specific intermediate outcomes. More explanation is needed for how the theory of change contributed to achieving the outcomes. A consideration of the limitations of this approach that the authors might have come across is also needed.

9. Reach. The study took place during the pandemic which was a unique context that led to increased student engagement with online learning more generally. Can the authors please comment on how that might have affected their results?

10. Conclusion. The conclusion refers to the theory of change as an intervention. If it is being conceptualised as an intervention, a case for this and an explanation should be provided in the introduction. The authors also conclude that the flexibility of the theory of change structure has allowed the organisation to reach its goals. What do the authors mean by flexibility? Do they really mean the interventions, rather than the theory of change itself?

11. Stakeholders (line 4 of the introduction) – please make clear stakeholders of what?

12. On first reference to LMIC please spell out what this is.

13. ‘The SRF core team realised they needed a methodology with an adequate level of operational details informed by the contextual needs to facilitate major reforms…’ (bottom of first page of introduction. I was not sure what was meant by this sentence. Please expand and explain.

14. Conceptualisation. Page numbers are needed for the direct quotations from Da Silva et al, as well as a date.

15. Formation of core group. The authors state in the text that the core group were medical students, but table 1 seems to indicate that these were people in senior positions with much experience. This is confusing, I would recommend the authors provide more explanation. As point 1 above, have all of them given their consent to be named in this way?

6. PLOS authors have the option to publish the peer review history of their article (what does this mean?). If published, this will include your full peer review and any attached files.

Reviewer #1: No

---

## [Author Response · Author response to Decision Letter 0]

11 Feb 2023

1. Thank you for the comment. The relevant information has been added to the manuscript under methods section “II. Study Design” page 7, paragraph 1. Ethical approval was sought from the Ethics Review Committee at the Aga Khan University and exemption was received (approval number 2022-7137-20351). The individuals named in this paper, along with the information on their involvement in the process, have provided their consent. Participant data is employed only in the form of numbers available from social media platforms or event attendance and is completely anonymized. 

2. We agree with the reviewer’s comment and have reframed the paper accordingly. We have changed the title to: ‘Implementation evaluation of a medical student-led intervention to enhance students’ engagement with research: findings and lessons learned.’ The theory of change has been reported under methods as the approach used to design the intervention activities.

The discussion has been enhanced to include limitations and potential benefits of the approach as experienced by the medical students leading the intervention.

3. We have elaborated the methods section on page 10, para 2 to explain the context of why certain interventions were chosen e.g., social media as an intervention to enhance faculty engagement and recognition. The specific intervention activities were designed after discussion and mutual agreement of the core members during the design phase. Being part of the college, the core group was very aware of what the reasons were for low engagement of the stakeholders. The main guiding principle was to incentivize engagement through social media and participation of higher leadership. 

4. We have described development of the theory of change under methods section “III. Development of ToC”, page 7. The process for the formation of a theory of change included weekly or biweekly meetings between January to June 2019, with each meeting lasting from 45 minutes to an hour. Meetings would involve updates on a task list e.g. the evaluation and improvement of organization objectives. The fulfillment of these objectives would then be viewed through problems and obstacles for which solutions needed to be created. Progress was recorded as meeting minutes that eventually coalesced to form a primitive ToC. In the following meetings, students in the core group would present a draft of theory of change. Feedback was provided by mentors through questioning about the suggested activities and why this would encourage greater engagement. Principles of the intervention activities were based on incentive, feasibility - free courses, capability of the students, student requirements, and the efficacy of whether these strategies would be effective or not.

5. Implementation outcomes were adapted from Proctor et. al, cited as reference 18, and are defined in the following paragraphs in the data analysis section. The outcomes were decided by the core group during the design based on principles of feasibility of collecting data for the students who had to manage it along with their ongoing classes.

6. Thank you for the comments. We have clarified in the results section that we describe the implementation evaluation of our intervention under page 27, paragraph 1. 

Two authors were designated for data collection (RA, AS). The process of data collection and analysis has been described under “V. Data Acquisition and Analysis” on page 16, paragraph 3.

7. Thank you for the comment. The figure has been modified as a logical framework to reflect the involvement of the interventions (listed as “activities”), steps in the process, and the desired long-term goals. The modified figure aims to accurately represent the study as a commentary on the success of the interventions instead of solely the theory of change.

8. We agree with the comment. We have revised the manuscript to describe how the approach using theory of change has helped the students (page 42, paragraph 2). 

9. The COVID pandemic may have facilitated participation through means of a flexible schedule, allowing more participants to join. An increase in laptop acquisition during the pandemic could have also contributed to increased participation (page 38, paragraph 2).

10. These interventions are effective, but partly due to the fact that these were informed by the theory of change. The ToC was designed by a multidisciplinary team and framed by a behavioral scientist with much deliberation on why the proposed intervention would work. We have rephrased the conclusion that the flexibility we refer to involves the interventions and not the theory of change itself. 

11. Stakeholders refers to medical students and has been modified appropriately.

12. Spelled out as Low and Middle Income Countries.

13. The SRF core team realized they needed a methodology with an adequate level of operational details to facilitate major reforms needed to meet the expanded scope. The team acknowledged the lack of a set protocol to follow (page 5, paragraph 1).

14. Page numbers and a date has been added for the quotation from Da Silva et. al.

15. The core group consisted of senior 3rd, 4th and 5th year medical students, along with faculty mentors with experience in implementation science, all of whom had been involved in research at some stage of their undergraduate medical career (Table 1). All individuals mentioned in table 1 have provided consent to be named with their roles and expertise.

---

## [Decision Letter · Decision Letter 1]

2 Aug 2023

PONE-D-22-11271R1Implementation evaluation of a medical student-led intervention to enhance students’ engagement with research: findings and lessons learnedPLOS ONE

Dear Dr. Haroon,

Thank you for submitting your manuscript to PLOS ONE. After careful consideration, we feel that it has merit but does not fully meet PLOS ONE’s publication criteria as it currently stands. Therefore, we invite you to submit a revised version of the manuscript that addresses the points raised during the review process.

We look forward to receiving your revised manuscript.

Kind regards,

Yolanda Malele-Kolisa, BDS, MPH, MDent, PhD

Academic Editor

PLOS ONE

Reviewers' comments:

Reviewer's Responses to Questions

**Comments to the Author**

1. If the authors have adequately addressed your comments raised in a previous round of review and you feel that this manuscript is now acceptable for publication, you may indicate that here to bypass the “Comments to the Author” section, enter your conflict of interest statement in the “Confidential to Editor” section, and submit your "Accept" recommendation.

Reviewer #1: All comments have been addressed

Reviewer #2: (No Response)

2. Is the manuscript technically sound, and do the data support the conclusions?

Reviewer #1: Yes

Reviewer #2: Partly

3. Has the statistical analysis been performed appropriately and rigorously? 

Reviewer #1: Yes

Reviewer #2: I Don't Know

4. Have the authors made all data underlying the findings in their manuscript fully available?

Reviewer #1: Yes

Reviewer #2: Yes

5. Is the manuscript presented in an intelligible fashion and written in standard English?

Reviewer #1: Yes

Reviewer #2: Yes

6. Review Comments to the Author

Reviewer #1: Thank you for submitting your revised paper. I am happy that you have addressed the comments and the paper is much improved.

Reviewer #2: Please update the ethics statement

It is not clear whether all the of 3rd to 5th years medical students were included in the formation of core groups. The authors did state that they selected those who had been involved in research but did not explain the extent of involvement (whether completed the research module, conducted research, had publication).

It is not clear how those who were involved at each stage, were recruited.

Number of participants benefited at each intervention is not clear. e.g., Indicate number of attendees where the number of workshops is reported.

Did you document the characteristics of participants who benefitted from each intervention?

Please include the denominator when reporting the outcomes of each intervention.

Who facilitated the SRF?

Did you have the same members in the core group for the total duration of the project?

7. PLOS authors have the option to publish the peer review history of their article (what does this mean?). If published, this will include your full peer review and any attached files.

Reviewer #1: No

Reviewer #2: No

---

## [Author Response · Author response to Decision Letter 1]

3 Aug 2023

August 3, 2023

Dear Editor and Reviewers,

Thank you for reviewing and critiquing our manuscript and for providing thorough comments to allow us to improve our paper. I have tabulated each comment below along with a response indicating how the authors have worked on it accordingly. Please let us know if any further revisions are required. Once again, we are grateful to the team at PLOS ONE for their consideration of our work. 

Warm regards,

Mian Arsam Haroon, MBBS

Aga Khan University

mian.haroon22@alumni.aku.edu

Comment Response

1. Please update the ethics statement 

The authors regretfully were not able to fully understand the reviewer’s comment. Based on the previous review and response to reviewers submitted by the authors, we have already updated the ethics statement as mentioned under methods section “II. Study Design” page 7, paragraph 1. The IRB approval for the project is attached as “ERC approval SRF ToC (1)”. Please let us know if any other changes are required.

2. It is not clear whether all the of 3rd to 5th years medical students were included in the formation of core groups. The authors did state that they selected those who had been involved in research but did not explain the extent of involvement (whether completed the research module, conducted research, had publication).

 We agree with the ambiguity pointed out by the reviewer. We have updated the manuscript to mention that only a few, not all, 3rd to 5th year medical students were involved in the formation of the core groups. We have also mentioned that the students had completed their research module and had conducted some sort of research till that point (page 8, paragraph 3).

3. It is not clear how those who were involved at each stage, were recruited. 

The authors have clarified in the revised manuscript that the participation of members in the core group was voluntary, given that they fulfilled the minimum requirements of research involvement in the comment #2, which all senior SRF members did (page 8, paragraph 3).

4. Number of participants benefited at each intervention is not clear. e.g., Indicate number of attendees where the number of workshops is reported. 

We agree with the reviewer’s comments. In the example mentioned, the authors have provided the number of attendees for certain workshops held by the Student Research Forum (page 30, paragraph 1). The number of attendees for all interventions are listed in the supporting file ‘S1 table’. Keeping in mind that the authors intended for this project to be a commentary on implementation science, for the sake of conciseness, we decided to include this information in the supporting file. If the reviewers would still prefer us to add the table to the main manuscript, the authors will gladly oblige. 

5. Did you document the characteristics of participants who benefitted from each intervention?

 Unfortunately, we were not able to document characteristics of the participants that benefitted in our interventions. We have clarified this on page 43 in paragraph 2. 

6. Please include the denominator when reporting the outcomes of each intervention. 

Although the authors understand the comment made by the reviewer, we regret that we are not able to provide the denominators for the outcomes. Per our viewpoint, the paper does not prioritize outcomes but aims to focus on the implementational interventions that facilitated the organization’s improvement. We see the unavailability of the denominator as twofold: 1) the denominator was too difficult to acquire e.g. the number of individuals reached out to sign up as research conference ambassadors or 2) not available e.g. the number of participants that signed up for an event but did not end up attending. 

7. Who facilitated the SRF? 

The SRF, with respect to this paper, was facilitated by the faculty patron Babar Hasan and global health faculty Muneera Rasheed. This is mentioned on page 8 in paragraph 2 and table 1. 

8. Did you have the same members in the core group for the total duration of the project? 

No. Over the course of this project, the number of members of the core group fluctuated. The number never went below the original 5 involved as mentioned in table 1. As members graduated from medical school, they would cease to be involved in the core group. As members were inducted into more senior roles in SRF, they would have the opportunity to be involved in the project. This has been clarified on page 13 in paragraph 1.

---

## [Decision Letter · Decision Letter 2]

18 Aug 2023

Implementation evaluation of a medical student-led intervention to enhance students’ engagement with research: findings and lessons learned

PONE-D-22-11271R2

Dear Dr. Haroon,

We’re pleased to inform you that your manuscript has been judged scientifically suitable for publication and will be formally accepted for publication once it meets all outstanding technical requirements.

Kind regards,

Yolanda Malele-Kolisa, BDS, MPH, MDent, PhD

Academic Editor

PLOS ONE

Additional Editor Comments (optional):

Thank you for submitting. Make sure the data availability statement is adhered to and submit the raw data as a supplementary file or according to the journal guidelines.

Reviewers' comments:

Reviewer's Responses to Questions

**Comments to the Author**

1. If the authors have adequately addressed your comments raised in a previous round of review and you feel that this manuscript is now acceptable for publication, you may indicate that here to bypass the “Comments to the Author” section, enter your conflict of interest statement in the “Confidential to Editor” section, and submit your "Accept" recommendation.

Reviewer #2: All comments have been addressed

2. Is the manuscript technically sound, and do the data support the conclusions?

Reviewer #2: Yes

3. Has the statistical analysis been performed appropriately and rigorously? 

Reviewer #2: N/A

4. Have the authors made all data underlying the findings in their manuscript fully available?

Reviewer #2: Yes

5. Is the manuscript presented in an intelligible fashion and written in standard English?

Reviewer #2: Yes

6. Review Comments to the Author

Reviewer #2: The authors have addressed all my comments and I'm happy with the responses on the query I had relating to the scientific presentation of results.

7. PLOS authors have the option to publish the peer review history of their article (what does this mean?). If published, this will include your full peer review and any attached files.

Reviewer #2: No

---

## [Editor Report · Acceptance letter]

23 Aug 2023

PONE-D-22-11271R2 

Implementation evaluation of a medical student-led intervention to enhance students’ engagement with research: findings and lessons learned 

Dear Dr. Haroon:

I'm pleased to inform you that your manuscript has been deemed suitable for publication in PLOS ONE. Congratulations! Your manuscript is now with our production department. 

Kind regards, 

on behalf of

Dr. Yolanda Malele-Kolisa 

Academic Editor

PLOS ONE